# Towards Activated Muscle Group Estimation in the Wild

## ABSTRACT

In this paper, we tackle the new task of video-based **A**ctivated **M**uscle **G**roup **E**stimation (AMGE) aiming at identifying active muscle regions during physical activity in the wild. To this intent, we provide the MuscleMap dataset featuring >15$K$ video clips with 135 different activities and 20 labeled muscle groups. This dataset opens the vistas to multiple video-based applications in sports and rehabilitation medicine under flexible environment constraints. The proposed MuscleMap dataset is constructed with YouTube videos, specifically targeting High-Intensity Interval Training (HIIT) physical exercise in the wild. To make the AMGE model applicable in real-life situations, it is crucial to ensure that the model can generalize well to numerous types of physical activities not present during training and involving new combinations of activated muscles. To achieve this, our benchmark also covers an evaluation setting where the model is exposed to activity types excluded from the training set. Our experiments reveal that the generalizability of existing architectures adapted for the AMGE task remains a challenge. Therefore, we also propose a new approach, TRANSM³E, which employs a multi-modality feature fusion mechanism between both the video transformer model and the skeleton-based graph convolution model with novel cross-modal knowledge distillation executed on multi-classification tokens. The proposed method surpasses all popular video classification models when dealing with both, previously seen and new types of physical activities. The contributed dataset and code will be publicly available.

## CCS CONCEPTS

• **Computing methodologies → Activity recognition and understanding**; **Scene understanding**.

## KEYWORDS

Activate muscle group estimation, activity understanding, scene understanding.

## 1 INTRODUCTION

Human activity understanding is important as it enables the development of applications and systems that can enhance healthcare, improve security, and optimize various aspects of daily life by automatically identifying and understanding human actions and behaviors [1, 30, 39, 68, 71, 89]. Knowing which skeletal muscles of the human body are activated benefits human activity understanding, and sport and rehabilitation medicine from multiple perspectives

*ACM MM, 2024, Melbourne, Australia*

© 2024 Copyright held by the owner/author(s). Publication rights licensed to ACM.
ACM ISBN 978-x-xxxx-xxxx-x/YY/MM
https://doi.org/10.1145/nnnnnnn.nnnnnnn

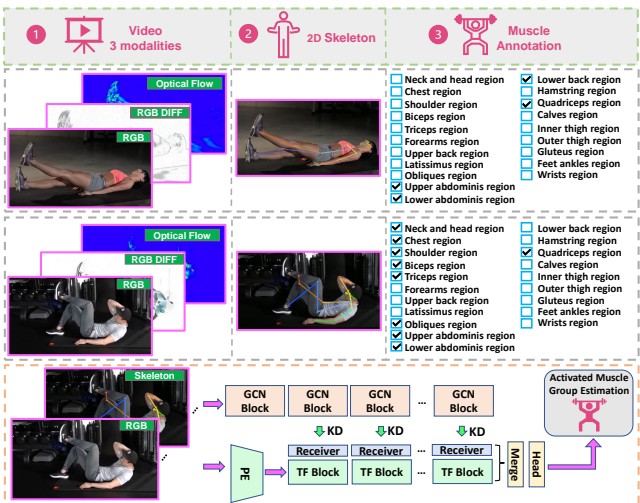

**Figure 1: Overview of the proposed MuscleMap dataset (Top) and the TRANSM³E model (Bottom). Our dataset contains four data modalities, *i.e.*, RGB, RGB difference (RGB Diff), optical flow, and 2D skeleton. PE and TF denote the patch embedding layer and the transformer block, respectively.**

and prevents inappropriate muscle usage which may cause physical injuries [20]. In health care, patients need to know how to conduct the exercise correctly to recover from surgery [35] or specific diseases [3], *e.g.*, COVID-19 [58]. Knowledge about muscle activations allows for user-centric fitness applications providing insights for everyday users or professional athletes who need specially adapted training. The majority of existing work on Activated Muscle Group Estimation (AMGE) is based on wearable devices with electrode sensors [15]. Yet, many wearable devices are inconvenient and heavy [41], even harmful to health [4], and have limited usage time due to the battery [64]. A big strength of wearable devices is the high accuracy achieved through direct signal measurement from skin or muscle tissue. However, such exact bio-electrical changes are not required in a large number of medical recovery programs, and knowing the binary activation status of the muscle as shown in Figure 1 is sufficient in many situations [47, 61, 78]. In contrast to wearable devices, most people have a video camera available at hand on their phone or laptop. Applying video-based AMGE on in-the-wild data collected by using smartphones or other widely available smart devices would allow for the application of such programs even without access to specialized hardware. Thereby, end-to-end video-based AMGE approaches are expected to be developed to prevent overburdens caused by wearable devices from both physical and psychological points of view. *Can modern deep learning algorithms relate fine-grained physical movements to individual muscles?* To answer this question, we tackle the barely researched task of video-based active muscle group estimation under an in-the-wild setting, which estimates muscle contraction during physical activities from video recordings without a restricted environment and background constraints.

Current research in video-based AMGE is limited by small-scale datasets and constrained data collection settings [13], where the data is often annotated with sensor signals and confined to restricted environments, covering only a limited range of actions. However, with the expansion of deep learning model capacities, there is a pressing need for larger datasets encompassing a wider variety of environments and activities. This expansion is vital for advancing the field of video-based AMGE within the research community.

In this work, we collect the first large-scale in-the-wild AMGE dataset from YouTube without environment constraints and give binary activation for different muscle regions by inquiring about sports field researchers. We created the MuscleMap dataset — a video-based dataset with 135 different exercises collected from YouTube considering in-the-wild videos. Each exercise type is annotated with one or multiple out of 20 different muscle group activations, as described in Table 1, which opens the door for video-based activated muscle group estimation in the wild task to the community. We annotate the dataset in a multi-label manner since human body movement is produced by the coordinated operation of diverse muscle regions. To acquire such annotations, we ask two senior researchers in the biomedical and sports research field to give the annotations.

We select various off-the-shelf Convolutional Neural Networks (CNNs) [8, 23], Graph Convolutional Networks (GCNs) [11, 36, 81], and transformer-based architectures [21, 40, 45] from human activity recognition field together with statistic methods as baselines. Our proposed MuscleMap benchmark describes a multi-label classification problem where each sample might be annotated with one or up to twenty labels. However, we find that it is challenging for all these models when they deal with new activity types containing new activated muscle combinations at test time considering the AMGE generalizability. Skeleton-based models are observed to show good performance on the new activity types while working not well on known activity types. The video-based models show good performance on the known activity types while delivering limited performance on the new activity types. An approach that can work well on both known and new activity types is thereby expected.

To tackle the aforementioned issue, we propose TRANSM$^3$E, a cross-modality knowledge distillation and fusion architecture that combines RGB and skeleton data via a new classification tokens-based knowledge distillation and fusion mechanism. To achieve better extraction of underlying cues for AMGE, we propose and equip TRANSM$^3$E with three essential novel components, *i.e.*, *Multi-Classification Tokens (MCT)*, *Multi-Classification Tokens Knowledge Distillation (MCTKD)*, and *Multi-Classification Tokens Fusion (MCTF)*, atop the most competitive performing architecture MViTv2 [40] as the backbone. As it is fundamental to mine and predict the activities at the global level for AMGE, the proposed TRANSM$^3$E, appearing as a transformer-based approach, is endowed with the capacity for long-term reasoning of visual transformers [76]. Since AMGE is a multi-label classification task, MCT is introduced, in view that using more classification tokens is expected to introduce more benefits toward finding informative cues. MCT also builds up the base for cross-modality MCT-level knowledge distillation.

Knowledge distillation [29] is leveraged for cross-modality knowledge transfer after the feature map reduction of the transformer block to enable a more informative latent space learning for different modalities. Transferring cross-modality knowledge during training

**Table 1: A comparison among the statistics of the video-based datasets, where AR, AQA, and CE indicate activity recognition, activity quality assessment, and calorie consumption estimation.**

| Dataset | NumClips | Task | MultiLabel | NumActions |
|---|---|---|---|---|
| KTH [32] | 599 | AR | False | 6 |
| UCF101 [69] | 13,320 | AR | False | 101 |
| HMDB51 [34] | 6,849 | AR | False | 51 |
| ActivityNet [7] | 28,108 | AR | False | 200 |
| Kinetics400 [7] | 429,256 | AR | False | 400 |
| Video2Burn [53] | 9,789 | CE | False | 72 |
| MTL-AQA [51] | 1,412 | AQA | True | / |
| FineDive [79] | 3,000 | AQA | True | 29 |
| FineGym [65] | 32,697 | AQA | True | 530 |
| MiA [13] | 15,000 | AMGE | False | 15 |
| MuscleMap135 (Ours) | 15,004 | AMGE | True | 135 |

significantly benefits the model in finding out cross-modality informative cues for the AMGE task. However, we find that it is difficult to achieve the knowledge distillation between two models with obvious architecture differences, *e.g.*, graph convolutional networks (GCNs) and video transformers, considering the alignment of the feature maps coming from different backbones and modalities to achieve the appropriate and effective knowledge distillation. Aside from the architectural differences, we examine that using late fusion to fuse the skeleton-based model and video-based model can not achieve a satisfactory performance due to the lack of alignment of the two different feature domains.

We thereby propose a cross-modality MCT-level knowledge distillation scheme considering knowledge distillation on the intermediate and final layers by specifically designing a knowledge distillation MCT for the model of each modality. Alongside the MCT used solely for the classification, we leverage another MCT to execute the knowledge distillation for each modality. The cross-modality knowledge distillation is then executed only between the knowledge distillation MCTs from the two modalities, whereas existing works mostly use knowledge distillation calculated from the full embeddings or single knowledge distillation token at the final layer and use a larger teacher [22, 29, 42, 48, 75]. While the mentioned MC-TKD mechanism integrates cross-modal knowledge into our main network with additional MCT for the knowledge distillation, another contribution, MCTF, merges the MCT of the distilled knowledge and the MCT for classification to achieve a final prediction towards the active muscle regions during human body motion for each modality. By combining these three components, TRANSM$^3$E achieves state-of-the-art performances with superior generalizability compared to the tested baselines.

In summary, our contributions are listed as follows:

- We open the vistas of video-based Activated Muscle Group Estimation in the wild task with the aim of lowering the threshold of entry to muscle-activation-based health care and sports applications.
- We provide a new benchmark MuscleMap to propel research on the aforementioned task which includes the large-scale *MuscleMap* dataset. We also present a large number of baseline experiments for this benchmark, including CNN-, transformer-, and GCN-based approaches.

- We especially take the evaluation of the generalizability into consideration by constructing test and validation sets using new activities excluded during the training.
- We propose TRANSM³E, targeting improving the AMGE generalizability towards new activity types. *Multi-classification Tokens* (MCT), *Multi-Classification Tokens Knowledge Distillation (MCTKD)* and *Multi-Classification Tokens Fusion (MCTF)* are used to formulate TRANSM³E, which shows superior generalizability on new activities and introduces state-of-the-art results on the MuscleMap benchmark.

## 2 RELATED WORK

**Activate Muscle Group Estimation (AMGE)** analysis is predominantly performed using electromyographic (EMG) data [5, 74] either with intramuscular (iEMG) or surface EMG sensors (sEMG). These methods use EMG data as input and detect activated muscle groups to achieve an understanding of the human body movement and the action, while we intend to infer muscle activations from body movements, therefore describing the opposite task. Chiquier *et al.* [13] propose a video-based AMGE dataset by using the signal of the wearable devices as the annotation. Yet, the data collection setting and the environment are restricted. The scale of the introduced dataset is relatively small and it encompasses limited action types. In our work, we collect a large-scale dataset based on HIIT exercises on YouTube while delivering binary annotation for each muscle region. We reformulate it into a multi-label classification task, namely AMGE in the wild. The annotations are first derived from online resources and then checked and corrected by researchers in sports fields.

**Activity Recognition** is a dominating field within visual human motion analysis [1, 10, 28, 30, 39, 68, 72, 85, 89] which was propelled by the advent of Convolutional Neural Networks (CNNs) with 2D-CNNs [25] in combination with recurrent neural networks (RNNS) [17] or different variations of 3D-CNNs [8, 23, 54]. More recently, transformer-based methods advanced over 3D-CNNs, especially with advanced pre-training methods and large datasets [40, 44, 45]. Action Quality Assessment (AQA) [51, 73] and Visual Calory Estimation (VCE) [53] relate to our work since these methods likewise shift the question of research from *what?* to *how?* with the aim of detailed analysis of human motion. Multimodal data is a common strategy, *e.g.*, by combining RGB video with audio [2, 52, 57], poses [62], optical flow [57], or temporal difference images [50]. Skeleton data is also commonly used as a modality for activity recognition on their own. Yan *et al.* [81] and follow-up research [11, 36, 66, 67] make use of GCNs, while competitive approaches leverage CNNs with special pre-processing methods [14, 19].

**Knowledge distillation (KD)** [29] became a common technique to reduce the size of a neural network while maintaining performance. In review [27], methods can be categorized to focus on knowledge distillation based on final network outputs (response-based) [31, 87], based on intermediate features (feature-based) [82, 86], or based on knowledge about the relations of data samples or features (relation-based) [9]. Recently, adaptations of distillation for transformer architectures gained attraction [37, 42]. Fusion strategies can be grouped into feature-fusion [56] and score fusion [33].

**Multi-label classification** methods allow for assigning more than a single class to a data sample. Common strategies include per-class binary classifiers with adapted loss functions to counter the imbalance problem [59], methods that make use of spatial knowledge [83, 84], methods that make use of knowledge about label relations [12, 70], or methods based on word embeddings [43, 80].

**Datasets** which combine visual data of the human body with muscle activation information are sparse and mainly limited to specific subregions of the human body, *e.g.*, for hand gesture recognition [26]. In contrast, a large variety of full-body human activity recognition datasets were collected in recent years, which are labelled with high-level human activities [34, 55], fine grained human action segments [38, 88], or action quality annotations [65]. We leverage such datasets by extending them with muscle group activation labels.

## 3 BENCHMARK

### 3.1 MuscleMap Dataset

With the new video-based active muscle group estimation in-the-wild task in mind, we collect the MuscleMap dataset by querying YouTube for the physical exercise video series. The collected dataset contains 135 activity types as well as 15,004 video clips and is competitive compared to other video-based datasets targeting fine-grained tasks, as shown in Table 1. Twenty activities are reserved for the validation and test splits of new activities, which are not included in the training set. MuscleMap targets physical exercise videos from fitness enthusiasts. High-Intensity Interval Training (HIIT) exercises are well suited for the AMGE in-the-wild task since they display a large range of motions that are designed to activate specific muscle groups and instructional videos provide high-quality examples of the displayed motion. The collected videos in our dataset are mostly near-person, which can benefit video-based muscle contribution understanding for the in-the-wild videos. A small set of activities from the MuscleMap dataset is shown in the bottom part of Figure 2. In Table 1, MuscleMap is compared with existing human activity recognition, action quality assessment, calorie consumption datasets, and time series-wise muscle activation regression dataset.

### 3.2 Activated Muscle Group Annotation

We cluster skeletal muscles of the human body into 20 major muscle groups with binary activation as shown in the checkboxes in Figure 1. To ensure the quality of the annotation, we ask 2 researchers from the biomedical and sports fields to give the annotation for each activity by watching the video from the dataset. If the two biomedical and sports researchers fully agree with the AMGE annotation towards one activity, this activity is included in our dataset. Both of the two annotators are senior researchers in the biomedical and sports fields.

### 3.3 Evaluation Protocol

To evaluate the generalizability of the leveraged approaches for the AMGE in-the-wild task, we formulate the **new val/test** and **known val/test**, where we use val and test to indicate the validation set and the test set, respectively. For MuscleMap, 20 of 135 activities are leveraged to formulate the **new val/test** set, which are *hollow hold, v-ups, calf raise hold, modified scissors, scissors, reverse crunches, march twists, hops on the spot, up and down planks, diamond push ups, running, plank jacks, archer push ups, front kicks, triceps dip*

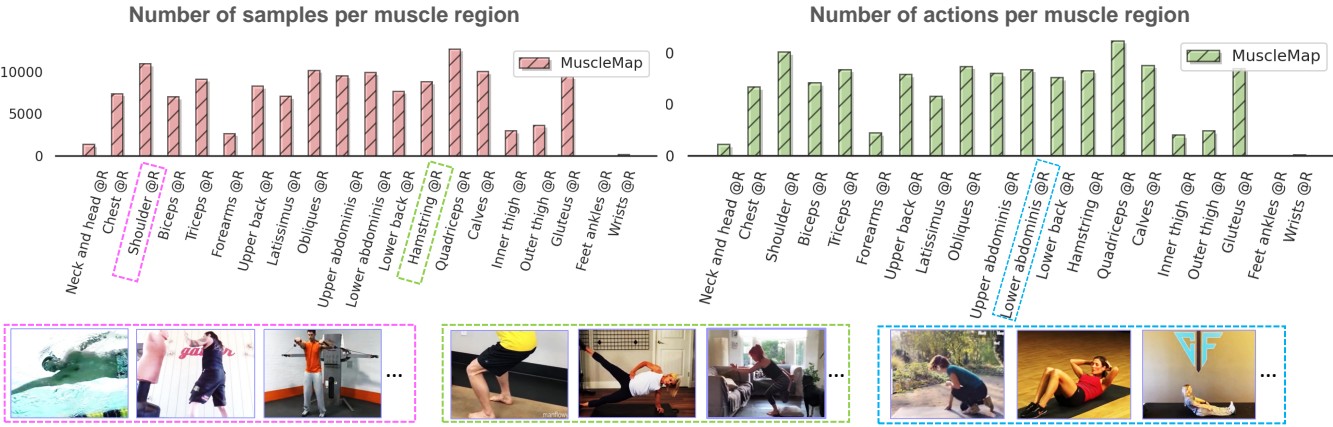

**Figure 2: An overview of the number of samples and the number of activity types per muscle region (@R), depicted at the top left and the top right. On the bottom, some activity-specific samples from the MuscleMap dataset are shown according to the corresponding muscle group.**

*hold, side plank rotation, raised leg push ups, reverse plank kicks, circle push ups, and shoulder taps*. The activity types for the **known test** and **known val** are the same as the activity types in the training set. The sample number for train, **new val**, **known val**, **new test**, **known test** sets are 7, 069, 2, 355, 1, 599, 2, 360, and 1, 594. The performances are finally averaged for new and known sets (**mean test** and **mean val**). We randomly pick up half of the samples from each **new** activity type to construct the **new val** while the rest of the samples from the selected **new** activities are leveraged to construct the **new test**. After the training of the leveraged model, we test the performance of the trained model on **known**/**new** evaluation and **known**/**new** test sets, and then average the performance of **known** and **new** sets to get the averaged performance on evaluation and test sets by considering both **known** and **new** activities which are both important for the AMGE in-the-wild task.

### 3.4 Evaluation Metric

Mean averaged precision (mAP) is used as the evaluation metric for the AMGE in-the-wild task. We let $\mathbf{l} = \{l_i | i \in [1, \ldots, N_l]\}$ denote the multi-hot annotation for the sample $i$ and $\mathbf{y} = \{y_i | i \in [1, \ldots, N_l]\}$ denote the prediction of the model for the given sample $i$. We first select the subset of $\mathbf{y}$ and $\mathbf{l}$ by calculating the mask through $\mathbf{m} = where(\mathbf{l} = 1)$. The corresponding subsets are thereby denoted as $\mathbf{y}[\mathbf{m}]$ and $\mathbf{l}[\mathbf{m}]$. Then we calculated the mean averaged precision score using the function and code from sklearn [6].

### 4 ARCHITECTURE

### 4.1 Preliminaries of MViT

TRANSM³E is based on the improved multi-scale visual transformer (MViTv2) [40], which is based on MViTv1 [21]. The model architecture of TRANSM³E is shown in Figure 3. Compared with ViT [18], MViTv1 increases the channel resolution progressively and reduces the resolution on the spatiotemporal plane simultaneously, which realizes pooling operations both on Keys (**K**) and Queries (**Q**). The basic idea of MViTv1 is the construction of different low- and

high-level visual modeling stages [21]. Multi-scale pooling attention is one of the major components of MViTv2 compared with ViT. MViTv2 uses decomposed relative position embeddings and residual pooling connections to integrate the principle of shift-invariance into the model and reduce computational complexity, while the downscaling in MViTv1 is achieved by large strides on the Keys (**K**) and Values (**V**).

### 4.2 Multi-Classification Tokens (MCT)

MCTs are used to harvest more informative components to achieve good generalizability for AMGE and to construct sender and receiver for cross-modality knowledge distillation in our work as shown in Figure 3. In our MCT setting, we directly use the final layer output of MCT and aggregate the MCT along the token dimension together with SoftMax to achieve multi-label classification.

Assuming the classification tokens of MCT to be referred to by $\{\mathbf{cls}_j | j \in [1, \ldots, C]\}$ and the flattened patch embeddings to be referred to as $\{\mathbf{p}_i | i \in [1, \ldots, N_{Patches}]\}$ for the given input video, where $N_{Patches}$ is the length of the patch sequence, the input of the first MViTv2 block is $[\mathbf{cls}_1, \ldots, \mathbf{cls}_C, \mathbf{p}_1, \ldots, \mathbf{p}_{N_{Patches}}]$. The final prediction $\mathbf{y}$ is computed through,

$$\mathbf{y} = SoftMax(\mathbf{P}_\alpha(\sum_{i=1}^{C} \mathbf{cls}_i / C), dim = -1), \quad (1)$$

where $\mathbf{P}_\alpha$ indicates a fully connected (FC) layer projecting the merged MCT to a single vector with the number of muscle regions as dimensionality. We make use of the same MCT settings for both the video-based backbone and the skeleton-based backbone according to Figure 4. After the first GCN block, the MCT for knowledge distillation and the MCT for classification are added to the model. We first flatten the spatial temporal nodes from the graph structure preserved by the GCN block. We use $\mathbf{z}^*_{GCN}$ to denote the nodes of the constructed graph structure, $\mathbf{cls}^*_m$ to denote the MCT for classification, and $\mathbf{cls}^*_r$ to denote the MCT for knowledge distillation regarding skeleton branch. We then concatenate all of these components along the node dimension and execute feature projection by using linear

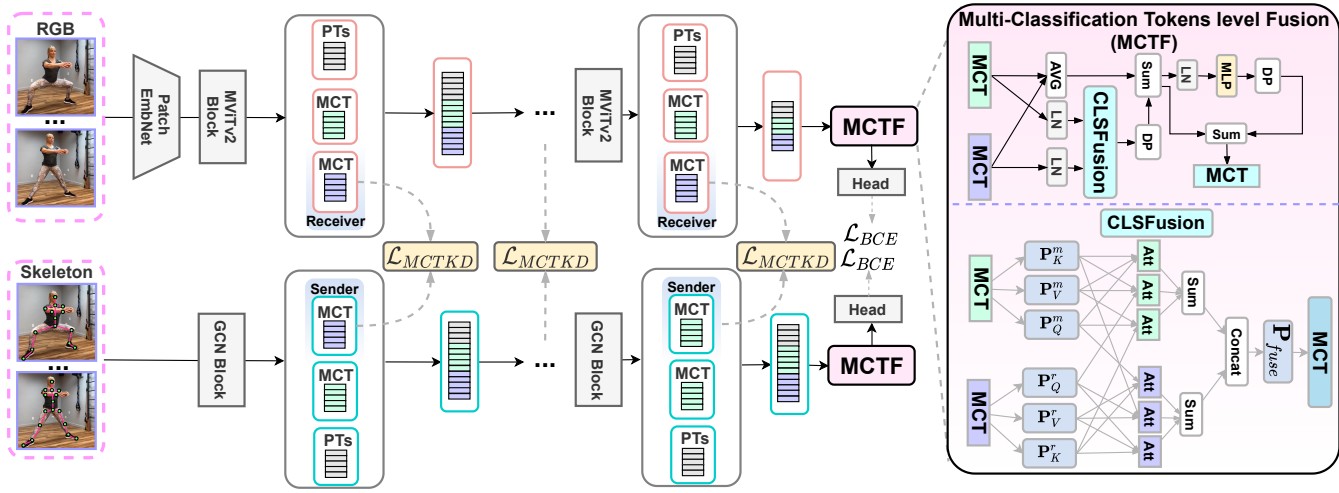

**Figure 3: TRANSM³E.** The knowledge is distilled between the skeleton branch and the RGB branch, PTs denote the patch embedding tokens. Three main components are shown, *e.g.*, Multi-Classification Tokens (MCTs), Multi-Classification Tokens Knowledge Distillation (MCTKD), and Multi-Classification Tokens Fusion (MCTF).

projection layer $\mathbf{P}_m$ as follows,

$$\mathbf{z}_{GCN}^*, \mathbf{cls}_m^*, \mathbf{cls}_r^* = Split(\mathbf{P}_m(Concate(\mathbf{z}_{GCN}^*, \mathbf{cls}_m^*, \mathbf{cls}_r^*))). \quad (2)$$

Then we execute an internal knowledge merge from the nodes to the MCT for the classification, as follows,

$$\hat{\mathbf{cls}}_m = \mathbf{P}_s(\mathbf{z}_{GCN}^*) + \mathbf{cls}_m^*, \quad (3)$$

where $\mathbf{P}_s$ denotes a FC layer. Finally, the node features, MCT for classification, and MCT for the knowledge distillation will be transferred to the next GCN block and the same procedure will be executed.

## 4.3 Multi-Classification Tokens Knowledge Distillation (MCTKD)

Multi-Classification Tokens Knowledge Distillation (MCTKD) is one of our main contributions. To the best of our knowledge, we are the first to introduce this technique which can enable knowledge distillation on the multi-classification tokens between two backbones with obvious structure differences. Through our observation in this work, we find that directly merging the feature from the skeleton-based model and video-based model can not achieve a satisfactory performance due to the huge structure and modality difference. To achieve appropriate feature fusion between two architectures on different modalities with obvious differences, we need a new solution focusing on this issue. Knowledge distillation, which has the capability to accomplish the feature space alignment, is firstly explored in our work on the MCT perspective to assist the cross-modality feature fusion for the AMGE in-the-wild task.

In the past, transformer-based knowledge distillation mainly focused on using intermediate full patch embeddings [48] or final classification token [75], while we propose knowledge distillation on the proposed MCT for both intermediate and final layers by using additional MCT for the knowledge distillation.

The underlying benefit of MCTKD is that the token number of the MCT is fixed, while knowledge distillation on the patch embeddings [22] may encounter the alignment issue when facing different

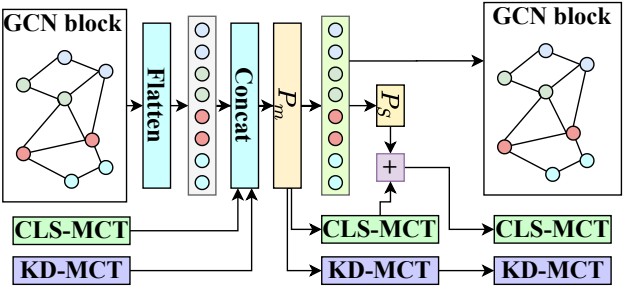

**Figure 4: An overview of the modified GCN block with knowledge distillation MCT and classification MCT.**

modalities with different token sizes. Instead of directly distilling knowledge from the MCT of an auxiliary modality towards the MCT of a major modality, knowledge distillation MCT is introduced to serve as a knowledge receiver. This approach avoids disruption on the MCT for classification for the major modality, *i.e.*, RGB video modality. The knowledge distillation MCT of the major modality branch is denoted as $\mathbf{cls}_r = \{\mathbf{cls}_{r,1}, \mathbf{cls}_{r,2}, ..., \mathbf{cls}_{r,C}\}$ and the knowledge distillation MCT from the branch of auxiliary modality is indicated by $\mathbf{cls}_s = \{\mathbf{cls}_{s,1}, \mathbf{cls}_{s,2}, ..., \mathbf{cls}_{s,C}\}$, MCTKD is achieved by applying KL-Divergence (KL-Div) loss after each feature map reduction block of MViTv2 on $\mathbf{cls}_r$ and $\mathbf{cls}_s$:

$$L_{MCTKD,all} = (\sum_{i=1}^{N_B} KL\text{-}Div(\mathbf{cls}_r^i, \mathbf{cls}_s^i))/N_B, \quad (4)$$

where $N_B$ and $L_{MCTKD,all}$ refer to the block number and the sum of MCTKD losses. $L_{MCTKD,all}$ is combined equally with the binary cross entropy loss ($L_{BCE}$).

## 4.4 Multi-Classification Tokens Fusion (MCTF)

Multi-Classification Tokens Fusion (MCTF) is designed to fuse MCT for knowledge distillation and the MCT for classification as in

Figure 3. We use $\mathbf{cls}_r$ to denote the knowledge distillation MCT, and $\mathbf{cls}_m$ denotes the classification MCT. $\mathbf{K}$, $\mathbf{Q}$, and $\mathbf{V}$ for each MCT can be obtained through linear projections $\mathbf{P}_K^m$, $\mathbf{P}_Q^m$, $\mathbf{P}_V^m$, $\mathbf{P}_K^r$, $\mathbf{P}_Q^r$, and $\mathbf{P}_V^r$ as follows,

$$\mathbf{K}_m, \mathbf{Q}_m, \mathbf{V}_m = \mathbf{P}_K^m(\mathbf{cls}_m), \mathbf{P}_Q^m(\mathbf{cls}_m), \mathbf{P}_V^m(\mathbf{cls}_m),$$
$$\mathbf{K}_r, \mathbf{Q}_r, \mathbf{V}_r = \mathbf{P}_K^r(\mathbf{cls}_r), \mathbf{P}_Q^r(\mathbf{cls}_r), \mathbf{P}_V^r(\mathbf{cls}_r). \tag{5}$$

After obtaining the $\mathbf{Q}_{\mathbf{m/r}}$, $\mathbf{K}_{\mathbf{m/r}}$, and, $\mathbf{V}_{\mathbf{m/r}}$ from the MCT for classification and the MCT for the knowledge distillation, a mixed attention mechanism is calculated as follows,

$$\mathbf{A}_{mm}^m = \mathbf{P}_{mm}(DP(Att(\mathbf{Q}_m, \mathbf{K}_m, \mathbf{V}_m))),$$
$$\mathbf{A}_{mr}^m = \mathbf{P}_{mr}(DP(Att(\mathbf{Q}_m, \mathbf{K}_r, \mathbf{V}_m))), \tag{6}$$
$$\mathbf{A}_{rm}^m = \mathbf{P}_{rm}(DP(Att(\mathbf{Q}_r, \mathbf{K}_m, \mathbf{V}_m))),$$

where $Att$ denotes the attention operation $Att(\mathbf{Q}_{\mathbf{m/r}}, \mathbf{K}_{\mathbf{m/r}}, \mathbf{V}_{\mathbf{m/r}}) = SoftMax(\mathbf{Q}_{\mathbf{m/r}}@\mathbf{K}_{\mathbf{m/r}}) * \mathbf{V}_{\mathbf{m/r}}$ and DP indicates Dropout. The above equations provide attention considering different perspectives including self-attention $\mathbf{A}_{mm}^m$ and two types of cross attention, *i.e.*, $\mathbf{A}_{rm}^m$ and $\mathbf{A}_{mr}^m$ which use the Queries from the MCT for the classification and the Keys from the MCT for knowledge distillation and vice versa. The same procedure is conducted for the knowledge distillation MCT to generate $\mathbf{A}_{rr}^r$, $\mathbf{A}_{rm}^r$, and $\mathbf{A}_{mr}^r$ with DP by,

$$\mathbf{A}_{rr}^r = \mathbf{P}_{rr}(DP(Att(\mathbf{Q}_r, \mathbf{K}_r, \mathbf{V}_r))),$$
$$\mathbf{A}_{rm}^r = \mathbf{P}_{rm}(DP(Att(\mathbf{Q}_r, \mathbf{K}_m, \mathbf{V}_r))), \tag{7}$$
$$\mathbf{A}_{mr}^r = \mathbf{P}_{mr}(DP(Att(\mathbf{Q}_m, \mathbf{K}_r, \mathbf{V}_r))).$$

Then the attention is finalized as,

$$\mathbf{A}_m = Sum(\mathbf{A}_{mm}^m, \mathbf{A}_{mr}^m, \mathbf{A}_{rm}^m),$$
$$\mathbf{A}_r = Sum(\mathbf{A}_{rr}^r, \mathbf{A}_{mr}^r, \mathbf{A}_{rm}^r). \tag{8}$$

The fused attention is thereby calculated through,

$$\mathbf{A}_f = \mathbf{P}_f(Concat(\mathbf{A}_m, \mathbf{A}_r)), \tag{9}$$

where $\mathbf{P}_f$ denotes an FC layer. The whole procedure is indicated by,

$$\mathbf{A}_f = CLS_f(LN(\mathbf{cls}_m), LN(\mathbf{cls}_r)), \tag{10}$$

where $LN$ demonstrates the layer normalization and $CLS_f$ is the CLS-Fusion. Assuming we use $\mathbf{cls}_a$ to denote the average of MCT for classification and the MCT for knowledge distillation by $\mathbf{cls}_a = (\mathbf{cls}_m + \mathbf{cls}_r)/2$, the final classification tokens are harvested by,

$$\mathbf{cls}_f = \mathbf{cls}_a + CLS_f(LN(\mathbf{cls}_r), LN(\mathbf{cls}_m)),$$
$$\mathbf{cls}_f := \mathbf{cls}_a + DP(\mathbf{M}_\theta(LN(\mathbf{cls}_f))), \tag{11}$$

where $\mathbf{M}_\theta$ denotes a Multi-Layer Perception (MLP) based projection and DP denoted dropout operation. MCTKD and MCTF are added after $N_{MCT}$ epochs of training of TRANSM$^3$E with only MCT, for both of the leveraged modalities and models. During the test phase, we make use of the average of the prediction results from the two branches as the final prediction.

## 4.5 Implementation Details

All the video models are pre-trained on ImageNet1K [16] using PyTorch 1.8.0 with four V100 GPUs. To reproduce TRANSM$^3$E, we first train MViTv2-S with only MCT for classification on RGB modality and HD-GCN with only MCT for classification on skeleton modality for 80 epochs and then train TRANSM$^3$E with all components for another 80 epochs. We use AdamW [46] with learning rate of $1e^{-4}$. The input video for *train*, *test*, and *val* is center cropped and rescaled as 224×224 with color jitter parameter as 0.4.

## 4.6 Analysis on the MuscleMap Benchmark

The results of different architectures on our benchmark are provided in Table 2. First, the approaches include *Random*, in which the muscle activation is predicted randomly, and *All Ones*, in which all the samples are predicted as using all the muscle regions. These two simple approaches are used to serve as statistic baselines. *Random* and *All Ones* show overall low performances with <30% mAP on all the evaluations. These statistical approaches are leveraged to make comparisons between deep-learning-based approaches to verify whether the model predicts muscle activation randomly or not. The skeleton-based approach, *e.g.*, HD-GCN [36], ST-GCN [81], and CTR-GCN [11], obviously outperform the statistic approaches and deliver promising performances when dealing with unseen activity types. Video-based approaches surpass statistic and skeleton baselines in terms of the AMGE of the known activities, where transformer-based approaches, *e.g.*, MViTv2 S/B [40] and VideoSwin S/B [45], and CNN-based approaches, *e.g.*, C2D [24], I3D [8], Slow [23], SlowFast [23], are leveraged. MViTv2-S shows good performance compared with the other methods due to the capability for reasoning long-term information and the multi-scale pooling setting which can extract informative cues from different abstract perspectives, especially with 79.6%, 79.7% for **mean val** and **mean test** on the MuscleMap dataset. However, we find that skeleton-based approaches work well on the new activities while they can not deliver satisfactory results on the known activities. On the other hand, video-based approaches work well on the known activities while they can not provide promising AMGE results on the new activities. A good AMGE model is expected to work well in both of the scenarios.

To simultaneously achieve good performances for both the new activities and the known activities, we would like to grasp the advantages both of the skeleton-based approaches and the video-based approaches. We proposed TRANSM$^3$E, which achieves feature fusion and knowledge distillation by using multi-classification tokens (MCT) on both the video-based model and the skeleton-based model, by using the most outperforming backbones from the two different modalities, *i.e.*, MViTv2-S and HD-GCN. This new proposed method incorporates knowledge distillation from the multi-classification token level and feature fusion from the multi-classification token level to harvest more underlying attributes of the body motion which can benefit the activated muscle group estimation task in the wild task. TRANSM$^3$E surpasses all the others by large margins. TRANSM$^3$E is a transformer-based approach due to the capability for long-term reasoning of visual transformers [76] since the AMGE should consider the activities at the global level, which requires long-term information reasoning. TRANSM$^3$E has 64.1%, 97.8%, 64.2%, and 81.0% mAP considering **new val**, **known val**, **new test**, and **known test** on our benchmark, while the generalizability to new activities is mostly highlighted. TRANSM$^3$E outperforms MViTv2-S by 1.4% and 1.3% on the **mean val** and **mean test**, which

**Table 2: Experimental results on the MuscleMap benchmark. Here, known val, new val, known test, and new test denote evaluation and test sets for normal and generalizable validation and test, respectively. mean val and mean test denote the averaged mean average precision (mAP) of normal and generalizable settings.**

| Model | #PM | known val | new val | mean val | known test | new test | mean test |
|---|---|---|---|---|---|---|---|
| | | | | | MuscleMap @ mAP | | |
| Random | 0.0M | 29.7 | 29.0 | 29.4 | 28.9 | 29.5 | 29.2 |
| All Ones | 0.0M | 28.2 | 28.1 | 28.2 | 27.8 | 28.6 | 28.2 |
| ST-GCN [81] | 2.6M | 90.4 | 63.5 | 77.0 | 90.5 | 63.3 | 76.9 |
| CTR-GCN [11] | 1.4M | 93.7 | 62.2 | 78.0 | 93.6 | 61.7 | 77.7 |
| HD-GCN [36] | 0.8M | 93.4 | 63.1 | 78.3 | 93.4 | 63.1 | 78.3 |
| C2D (R50) [24] | 23.5M | 97.2 | 59.1 | 78.2 | 97.4 | 58.5 | 78.0 |
| I3D (R50) [8] | 20.4M | 97.0 | 59.4 | 78.2 | 97.0 | 58.4 | 77.7 |
| Slow (R50) [23] | 24.3M | 96.8 | 60.7 | 78.8 | 96.9 | 60.5 | 78.7 |
| SlowFast (R50) [23] | 25.3M | 89.7 | 60.2 | 75.0 | 94.4 | 59.6 | 77.0 |
| MViTv2-S [40] | 34.2M | 97.7 | 61.4 | 79.6 | 97.9 | 61.4 | 79.7 |
| MViTv2-B [40] | 51.2M | 97.4 | 61.2 | 79.3 | 97.7 | 61.0 | 79.4 |
| VideoSwin-S [45] | 50.0M | 92.6 | 58.8 | 75.7 | 92.4 | 58.8 | 75.6 |
| VideoSwin-B [45] | 88.0M | 91.8 | 58.7 | 75.3 | 91.9 | 58.3 | 75.1 |
| TransM³E (Ours) | 55.4M | **97.8** | **64.1** | **81.0** | 97.8 | **64.2** | **81.0** |

**Table 3: Ablation for TransM³E on MuscleMap.**

| MCT | MCTKD | MCTF | known val | new val | mean val | known test | new test | mean test |
|---|---|---|---|---|---|---|---|---|
| | ✓ | ✓ | 95.7 | 62.1 | 78.9 | 95.9 | 62.0 | 79.0 |
| ✓ | | ✓ | 95.7 | 62.1 | 78.9 | 95.9 | 62.0 | 79.0 |
| ✓ | ✓ | | 95.4 | 62.4 | 78.9 | 95.6 | 62.1 | 78.9 |
| ✓ | ✓ | ✓ | **97.8** | **64.1** | **81.0** | **97.8** | **64.2** | **81.0** |

**Table 4: Ablation of MCTKD on MuscleMap.**

| Method | known val | new val | mean val | known test | new test | mean test |
|---|---|---|---|---|---|---|
| FL-KD | 96.5 | 63.0 | 79.8 | 96.4 | 63.4 | 79.9 |
| DE-KD | 95.9 | 63.9 | 79.9 | 96.6 | 63.9 | 80.3 |
| SP-KD | 97.5 | 63.0 | 80.3 | 96.7 | 63.1 | 79.9 |
| FL-MCTKD | 95.1 | 63.0 | 79.1 | 95.5 | 62.8 | 79.2 |
| DE-MCTKD | 95.1 | 63.3 | 79.2 | 95.2 | 63.4 | 79.3 |
| SP-MCTKD | **97.8** | **64.1** | **81.0** | **97.8** | **64.2** | **81.0** |

**Table 5: Ablation for the MCTF on MuscleMap.**

| Method | known val | new val | mean val | known test | new test | mean test |
|---|---|---|---|---|---|---|
| Sum [60] | 95.4 | 62.4 | 78.9 | 95.6 | 62.1 | 78.9 |
| Multiplication [60] | 94.5 | 62.8 | 78.7 | 94.7 | 62.8 | 78.8 |
| SelfAttention [49] | 97.4 | 62.9 | 80.2 | 97.6 | 62.8 | 80.2 |
| CrossAttention [49] | 94.9 | 63.7 | 79.3 | 95.1 | 63.5 | 79.3 |
| MCTF (ours) | **97.8** | **64.1** | **81.0** | **97.8** | **64.2** | **81.0** |

**Table 6: Comparison of MMF/KD on MuscleMap.**

| Method | known val | new val | mean val | known test | new test | mean test |
|---|---|---|---|---|---|---|
| LateFusionSum [60] | 80.6 | 59.8 | 70.2 | 80.1 | 60.0 | 70.1 |
| LateFusionConcat [77] | 83.5 | 60.8 | 72.2 | 83.3 | 61.2 | 72.3 |
| LateFusionMul [60] | 82.3 | 60.4 | 71.4 | 82.0 | 60.9 | 71.5 |
| Ours | **97.8** | **64.1** | **81.0** | **97.8** | **64.2** | **81.0** |

especially works well for new val and new test as TRANSM³E surpasses MViTv2-S by 2.7% and 2.8%. Our method achieves significant improvements in AMGE through three main strategies: enhanced attribute reasoning using MCT, effective knowledge exchange between video and skeleton models, and integration of distillation and classification MCTs for better information consolidation. Despite its strengths, the performance gap in AMGE between known and new activities highlights areas for improvement, particularly in reducing misclassifications and biases, with more insights to be shared in a forthcoming ablation study analysis.

## 4.7 Analysis of the Ablation Studies

**Module ablation.** The ablation study of MCT, MCTKD, and MCTF, is shown in Table 3, where we deliver the results for *w/o MCT*, *w/o MCTKD*, *w/o MCTF*, and *w/ all*. When we compare the results between *w/o MCT* and *w/ all*, we find that using MCT to enlarge the attributes prediction space can contribute performance improvements by 2.1%, 2.0%, 2.1%, 1.9%, 2.2%, and 2.0% in terms of known val, new val, mean val, known test, new test, and mean test. When comparing the results between *w/o MCTKD* and *w/ all*, we observe that leveraging MCTKD to achieve multi-stage information exchange from the video model and skeleton model can harvest performance improvements by 2.1%, 2.0%, 2.1%, 1.9%, 2.2%, and 2.0% in terms of the six aforementioned evaluations. When we compare the results between *w/o MCTF* and *w/ all*, we find that using MCTF to achieve the fusion between the information derived from the classification MCT and knowledge distillation MCT can bring performance improvements of 2.4%, 1.7%, 2.1%, 2.2%, 2.1%, and 2.1% in terms of the six aforementioned evaluations.

**MCTKD ablation.** We evaluate the effects of varying the location of knowledge distillation application and the knowledge distillation on a single distillation token (KD) and MCT (MCTKD), where they are named differently, *i.e.*, KD/MCTKD at the final layer (FL-KD/MCTKD), SP-KD/MCTKD after token size reduction (SP-KD/MCTKD), or KD/MCTKD after each MViTv2 block (DE-KD/MCTKD), in Table 4. Considering different knowledge distillation localizations, SP-KD and SP-MCTKD achieve the best performances for KD and MCTKD individually, demonstrating their superiority of using sparse knowledge distillation settings after the reduction of the feature map size. When we compare sparse knowledge distillation with dense knowledge distillation, SP-MCTKD achieves performance improvement by 2.7%, 0.8%, 1.8%, 2.6%, 0.8%, and 1.7% in terms of known val, new val, mean val, known test, new test, and mean test. When we compare sparse knowledge distillation with final layer knowledge distillation, SP-MCTKD achieves performance improvements by 2.7%, 1.1%, 1.9%, 2.3%, 1.4%, and 1.8% in terms of the aforementioned six evaluations.

Using MCTKD in sparse locations—specifically after each feature map size reduction—yields the best results on the MuscleMap benchmark. This method enhances the aggregation of AMGE cues across modalities after pooling, facilitating more effective knowledge distillation. SP-MCTKD surpasses SP-KD by utilizing multi-classification tokens to expand the prediction space for attributes during training, thereby capturing and transferring more crucial AMGE cues from various modalities. Consequently, SP-MCTKD, which excels in performance, is chosen as the foundation for our model, facilitating knowledge exchange between skeleton and video models. SP-MCTKD achieves the best performance and is selected.

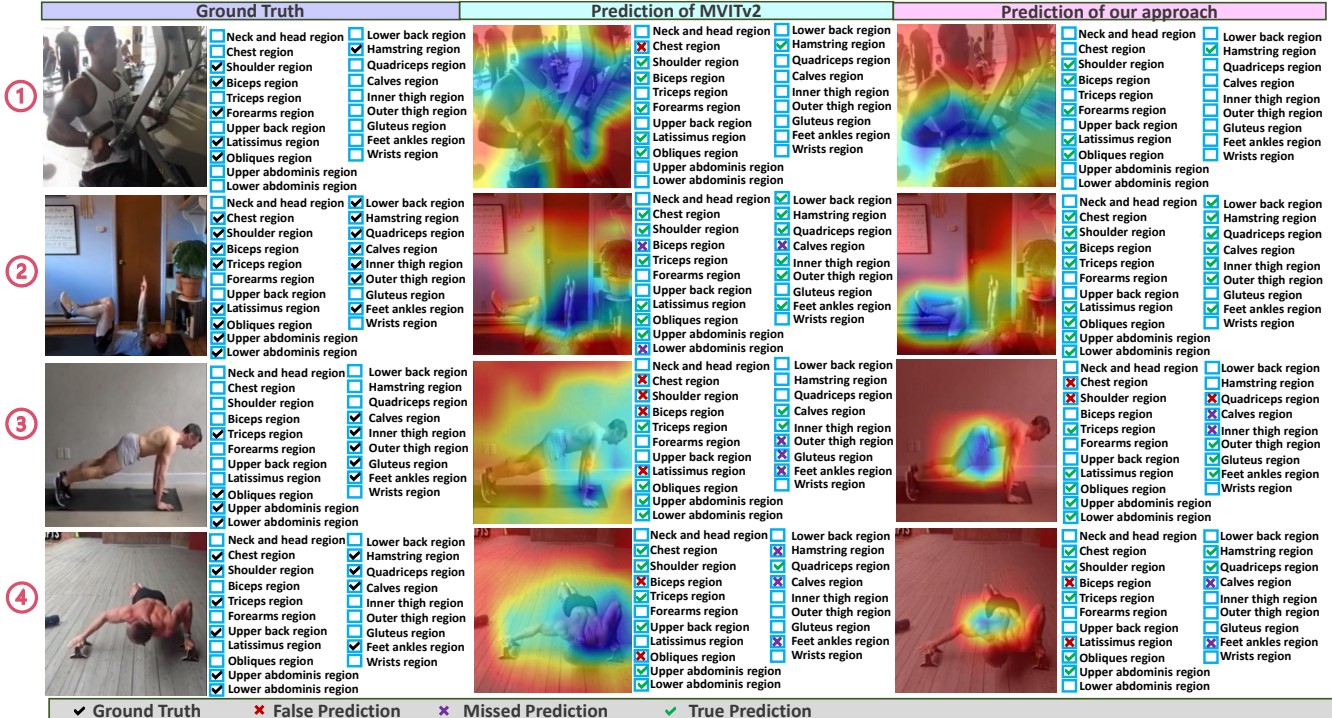

**Figure 5: Qualitative results for the MViTv2-S [40] and TRANSM³E. GradCam [63] visualization is given. The ground truth is shown on the left, and the prediction and gradients of the MViTv2-S [40], and our approach are shown in the middle and on the right.**

**MCTF ablation.** The ablations on MCTF for TRANSM³E are presented in Table 5, where our approach is compared with existing fusion approaches, *e.g.*, *Sum*, *Multiplication*, *SelfAttention*, and *CrossAttention*. MCTF shows the best performance with 81.0% and 81.0% on **mean val** and **mean test**. Compared with *CrossAttention*, our MCTF achieves performance improvement by 1.7% and 1.7% in terms of the **mean val** and **mean test**. The superiority of MCTF compared to other approaches, especially on generalizability, depends on using attention from a more diverse perspective, which benefits the capability of integration considering different focus formats.

## 4.8 Comparison with Conventional Multi-modality Fusion Approaches

Table 6 presents the comparison between TRANSM³E and existing multi-modality fusion approaches, *i.e.*, *LateFuionSum*, *LateFusionConcat*, and *LateFusionMul*. We compare our proposed method to these conventional multi-modal fusion approaches to illustrate that the performance improvement of our approach is not solely delivered by using the feature fusion between the skeleton modality and the RGB video modality. Compared with the best performing baseline *LateFusionConcat*, our approach achieves a performance improvement by 14.3%, 3.3%, 8.8%, 14.5%, 3.0%, and 8.7% in terms of **known val**, **new val**, **mean val**, **known test**, **new test**, and **mean test**.

## 4.9 Analysis of Qualitative Results

Qualitative results are shown in Figure 5, the label and GradCam [63] visualizations of MViTv2-S and TRANSM³E are given from left to right. The true/missed/false prediction is marked as green checkmark/purple crossmark/red crossmark. Overall, our approach has more accurate predictions and fewer false and missed predictions for all the samples considering known activities, *i.e.*, ① and ② in Figure 5, and new activities, *e.g.*, ③ and ④, where ① and ② are correctly predicted by our model. TRANSM³E concentrates mostly on the accurate body regions, *e.g.*, in sample ③ TRANSM³E focuses on the leg and abdominis related region, while the focus of the MViTv2-S is distracted, which results in more false predictions of MViTv2-S. Due to the integration of the learned knowledge from both the video and skeleton modalities, our model can achieve a better focus.

## 5 CONCLUSION

In this paper, we open the vistas of video-based activated muscle group estimation in the wild. We contribute the first large-scale video-based activated muscle group estimation dataset considering in-the-wild video and build up MuscleMap benchmark for the AMGE by using statistic baselines and existing video-based approaches including both video-based and skeleton-based methods. We take additional consideration regarding the AMGE generalizability. We propose TRANSM³E with multi-classification token distillation and fusion in a cross-modality manner to enhance the generalization to new activity types. TRANSM³E sets the state-of-the-art on the proposed MuscleMap benchmark and it delivers promising generalizability towards unseen activities in terms of the AMGE in-the-wild task. In the future, the utilization of large language models (LLMs) for the purpose of video-based active muscle group estimation is anticipated to enhance the extensibility of the approach with respect to estimating muscle groups during novel physical activities.

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
