# OpenReview forum: "Towards Video-based Activated Muscle Group Estimation in the Wild"
_acmmm.org/ACMMM/2024/Conference — MM2024 Poster_

### Official Review · Reviewer_oDnz · 2024-05-23

**Rating:** 3
**Confidence:** 4

**Summary:**

The authors introduce a new approach  $TRANSM^3E$ , utilizing a multi-modality feature fusion mechanism that combines Vision Transformer and GCN to process both RGB and Skeleton inputs. Additionally, the approach implements cross-modality knowledge distillation on multi-classification tokens.

**Strengths:**

- Traditional AMGE relies on sensors to obtain electromyogram signals. For video-based AMGE tasks, due to occlusions such as clothing or body parts, estimating activated muscle groups using intuitive human body motions is effective.

- The ablation studies in the paper are comprehensive.

**Limitations:**

- In Lines 136-137, the entire dataset annotation relies solely on two professional researchers, which raises concerns about potential cognitive biases. Additionally, there are unreasonable annotations in the GroundTruth part of Figure 5 (for example, is there no involvement of the chest muscle group in the shoulder movement in Sample 1?). Furthermore, the physical concept of "activation" in the task requires further clarification.

- Since the proposed MuscleMap dataset includes not only bodyweight exercises but also interactions with fitness equipment, the muscle activations in such interactive actions tend to differ from those in bodyweight resistance exercises, which is not discussed in the paper.

- The paper does not mention how the RGB Diff and optical flow modalities in the MuscleMap dataset are utilized, nor does it explain the acquisition method for the skeleton annotations.

- Input-output clarity: Does the proposed $TRANSM^3E$ output the recognition of action types and activated muscle groups for each input frame of RGB and skeleton?

- The backbone network MViTv2 and the compared work VideoSwin are both from before 2022, and the paper does not adequately discuss more recent work from the past year.

- In Section 4.6, the paper fails to effectively explain why skeleton-based methods and video-based methods cannot achieve good performance on both known actions and new emerging actions.

- Other: The paper title does not match the title on the OpenReview submission page.

**Suitability:**

3

---

### Official Review · Reviewer_G8gp · 2024-05-28

**Rating:** 5
**Confidence:** 4

**Summary:**

**summary**
this paper proposes a new task of muscle group activation, where for each activity, multi-labels of activated muscle groups are provided. a new method seems to perform reasonably well as compared to existing baselines like slowfast.

**Strengths:**

**strength**
authors propose:
- a new task of video-based activated-muscle group estimation which is a multi-label classification problem
- a new dataset/benchmark called MuscleMap, which consists of more muscle groups/activities
- Multi-Classification Tokens (MCT): classs-specific tokens are injected into the net, in contrast to vit where only one class token does into the net. this appears novel.

**Limitations:**

**weakness**

the weakness i mentioned below are some minor clarifications which authors can make, it does not impact the strong technical aspects presented in the paper.

**technical clarifications:**

- line 343: "For MuscleMap, 20 of 135 activities are leveraged to formulate the new val/test set": why are only 20 activities being considered in val/test. what about other 115? it would have made sense if these 20 were not seen during training, and zero-shot was being evaluated. however, it seems that this type of analysis was not in the scope of the paper?
however, i only see a softmax with fixed neurons being trained, not a contrastive thing as done in clip.

- line 505: "we find that directly merging the feature from the skeletonbased model and video-based model can not achieve a satisfactory
performance due to the huge structure and modality difference": is skeleton modality input as a sequence of coordinates, or black white skeleton map? this seems suprising, considering that there are other works like omni-bind which integrate multi-modality. is it possible to keep modality-specific encoder, but a SHARED backbone to process these two disparate modalities together? in that case, one might not need to process them with gcn layers.

- line 634- 642: it seems that the architecture is trained in 3 phases? any possible comments on making it  a joint -training instead?

- balanced dataset: it seems neck/head muscle group is lower in number... perhaps add some videos of people doing inverted headstand or other yoga pose involving head? for thigs, it could be a yogic pose like virbhadrasana?


**qualitative questions:**

- fig 5: shows gradcam on mvitv2 vs authors method. i see a blue blob at the correct place. but it is too broad, and not specifically focused on individual muscle group,
do the authors have any insights why this might be case?  perhaps some other form of qualitative analysis might be helpful.

**additional experiments:**

- an anlysis on open-world ability would be interesting: it might be tough given the rebuttal phase, but it definitely would be interesting, even if not, perhaps the authors could at least release an open-world TEST split so that the community could build on this work in the future.

- an analysis on "which" muscle group is toughest to classify? i think that muscles like neck etc which occupy less area might be tougher to do, but it would be interesting to see that being qualitatively analyzed, ...

- additional ablation: can the authors try one ablation with increasing some "filler" tokens, i.e. the parameters which dont contribute to the final softmax, but just used to guide the network to absorb some errors. does increasing such fillers results in better perfomance? [1]

**minor suggestions which dont impact the review**
 - line 223/912: please rephrase "the vistas of", sounds non-scientific, perhaps, "propose a new task of".

[1] hinton et al, how to representa part-whole hierarchies in connectionist-nets.

**rating**
weak accept.

**justification of rating**

this is a good work and i commend author's efforts in this work. It deserves acceptance PROVIDED that some of the clarifications, and additional experiments might be performed.... depending on how the rebuttal goes, i might **lower/retain my rating** later...

- thx,
your sincere reviewer :-)

**Suitability:**

3

---

### Official Review · Reviewer_s3N9 · 2024-05-29

**Rating:** 4
**Confidence:** 3

**Summary:**

This paper introduces a video-based activated muscle group estimation (AMGE) task, which aims to identify active muscle regions during physical activities in natural environments. To this end, the authors provide a dataset called MuscleMap, which includes over 15,000 video clips, covering 135 different activities and 20 labeled muscle groups. For practical application of the AMGE model, it is crucial to ensure that the model generalizes well to new types of activities and new combinations of activated muscles that were not present during training. The authors also propose a new method called TRANSM3E, which combines multimodal feature fusion between a video transformer model and a skeleton-based graph convolution model, along with cross-modal knowledge distillation, surpassing all popular video classification models.

**Strengths:**

1.The MuscleMap dataset is large-scale and unconstrained by the environment, which helps advance related research.
2.The proposed TRANSM3E method combines a video transformer and a skeleton graph convolution network, and uses multi-class tokens for cross-modal knowledge distillation, showing excellent performance and generalization ability.
3. This method and dataset have significant practical applications in fields such as sports and rehabilitation medicine.

**Limitations:**

1. The TRANSM3E model is quite complex, involving various techniques and components, which might increase implementation and deployment difficulties.
2. Despite the large dataset, all videos are from YouTube, which may have limitations in video quality and angles, affecting the model's generalization ability.
3. The dataset labeling relies on expert subjective judgment, which may lead to annotation errors and consistency issues. How can this bias be addressed?
4. Only one dataset, which is self-constructed, was used, making it hard to validate the algorithm's universality and robustness.

**Suitability:**

3

---

### Official Review · Reviewer_vRdo · 2024-05-30

**Rating:** 5
**Confidence:** 3

**Summary:**

This paper proposes a MuscleMap dataset for video-based Activated Muscle Group Estimation in the wild. It collects 135 different exercises from YouTube, each type of exercise is annotated with one or more of 20 different muscle group activations. There are always 15004 video sequences. This paper also proposes the $TRANSM^3E$ method to solve the Activated Muscle Group Estimation task of multi-label classification. It is a cross-modality knowledge distillation and fusion classification architecture, which includes three basic new components: Multi-Classification Tokens (MCT) (for multi-label tasks), Multi-Classification Tokens Knowledge Distillation (MCTKD), and Multi-Classification Tokens Fusion (MCTF). Firstly, MCT is used to process video-based models and backbone-based models. Then, MCTKD is used to fuse the multi-modality features of the two models, and an additional MCT is added to perform knowledge distillation on both the intermediate and final layers as a knowledge receiver. Finally, MCTF fuses the multi-classification labels used for knowledge distillation with those used for classification.

**Strengths:**

1. This paper proposes a new dataset called MuscleMap for the Activated Muscle Group Estimation task, which opens the prospect of video-based Activated Muscle Group Estimation and reduces the entry threshold for muscle activation-based healthcare and exercise applications.
2. The Multi-Classification Tokens Knowledge Distillation (MCTKD) proposed in this paper can fuse features of two backbone models with significant structural differences on different modalities, achieving feature space alignment.

**Limitations:**

1. The keywords are too large, broad, and not prominent enough.
2. The proposed dataset is all videos of professionals. Can you add some videos of ordinary people to verify the universality of the proposed dataset?
3. Inconsistent capitalization and abbreviations of paper titles in the references.

**Suitability:**

3

---

### Official Review · Reviewer_axRT · 2024-06-01

**Rating:** 5
**Confidence:** 3

**Summary:**

The paper explores a new task in the field of video-based Activated Muscle Group Estimation (AMGE). This task aims to identify active muscle regions during physical activities in various environments. The researchers introduce the MuscleMap dataset, collected from YouTube, specifically target High-Intensity Interval Training exercises, emphasizing real-world applications in sports and rehabilitation without stringent environmental constraints.

To address the challenges of this task, the researchers developed a new model called TRANSM3E. This model employs a feature fusion mechanism that combines video transformer models and skeleton-based graph convolution models. It also incorporates a novel approach to cross-modal knowledge distillation using multi-classification tokens. This innovative method allows the model to generalize effectively to new types of physical activities that were not included in the training data.

A significant contribution of this paper is the creation of the MuscleMap dataset, which provides a comprehensive and annotated video resource for AMGE. This dataset enables the development and benchmarking of AMGE models. The researchers conducted extensive experiments with various off-the-shelf models, including Convolutionsl Neural Networks, Graph Neural Networks, and transformer-based architectures, to evaluate their effectiveness in handling the AMGE task. These experiments revealed that generalizability remains a challenge for existing architectures adapted for AMGE. However, the proposed model demonstrated superior performance compared to popular video classification models, effectively handling both known and new activity types.

**Strengths:**

* Paper is well-structured on the higher level
* Paper introduces the novel task of **video-based** Activated Muscle Group Estimation (AMGE).
* Paper presents a novel dataset and benchmark
* The relevance for fitness apps or rehabilitation is described and it presents a lower entry barrier and less expensive approach compared to existing approaches
* Provides thorough evaluation using the newly created MuscleMap dataset, comparing the performance of TRANSM3E against various off-the-shelf models.
* Dataset includes a high number of clips and metadata
* Public available code and dataset

**Limitations:**

* The introduction contains much modeling work (i.e. 138-141; 141-143; 154-161, 202-207)
* The use of YouTube videos raises privacy and rights concerns, which are not addressed in the paper.
* No comparison with non-video-based methods as the mentioned wearable devices with electrode sensors, or real world/user study
* Title: based on the evaluation, is it a Towards paper?

**Suitability:**

2

---

### Official Review · Reviewer_8xcL · 2024-06-02

**Rating:** 4
**Confidence:** 2

**Summary:**

The paper introduces a  video-based solution for identifying active muscle regions during physical activity. Unless previous works, the proposed solution is designed for practical, non-controlled "in-the-wild" settings. The paper starts by first developping a rich datasets by exploting excercise YouTube videos and labelling them with the help of two researchers in the biomedical and sports research field. To address the  multi-label classification problem, the paper introduces TRANSM3E a cross-modality knowledge distillation and fusion architecture. TRANSM3E is thoroughly evaluated using the dataset collected from YouTube and is shown to outperform the state of the art, in particular in terms of generability.

**Strengths:**

- The dateset is very interesting and could provide very helpful to the commmunity.
- The idea of video-based physical activity recognition in non-controlled settings is very interesting and can have many practical applications.
- TRANSM3E is novel and shown to outperform the state of the art, especially in terms of generability, which is very important in practice.

**Limitations:**

- The writing and the structure need improvement to help with clarity. Section 4, for example, is called Architecture and yet it includes the performance evaluation as well. The writing is at times verbose and repetitive. For example, TRANSM3E is explained multiple times, including in the performance evaluation. A more focused and "lean" writing would help the reader better understand the contributions and the results.

**Suitability:**

3

---

### Official Review · Reviewer_qacp · 2024-06-02

**Rating:** 4
**Confidence:** 2

**Summary:**

This study tackles the challenge of Activity Muscle Group Estimation (AMGE), focusing on the identification of active muscle groups in the wild. The authors have created the MuscleMap dataset, an extensive resource comprising more than 15,000 video clips with 135 different activities and 20 muscle groups. This work introduces a method, TRANSM3E, which combines the strengths of video transformer model with skeleton-based graph convolutional model. The experimental results presented in the paper illustrate that TRANSM3E surpasses current video classification models in both its recognition of known activity types and its ability to generalize to new activities. It is commendable that the authors will release the source code and the MuscleMap dataset, which will facilitate further advancements in the field.

**Strengths:**

1.	The study introduces a task of AMGE in the wild, which presents prospects for applications in sports and the field of rehabilitation medicine.
2.	The construction of the MuscleMap dataset by the authors is commendable. Its large scale and diverse categories, coupled with the decision to make it publicly accessible, are likely to foster advancements in this area of research.
3.	The introduction of the TRANSM3E method is a contribution. The integration of transformer architectures and Multi-Class Tokens (MCT) is impressive, and the method's generalizability is evidenced by its state-of-the-art performance when benchmarked against leading video classification models.

**Limitations:**

1.	The dataset is currently limited to HIIT exercises, which somewhat narrows its scope for general use. To enhance its versatility, it would be advantageous to expand the dataset to include a broader range of activities, such as aerobic exercises and strength training.
2.	Section 3 lacks a detailed description of the methodology used to construct the skeleton data within the MuscleMap dataset. Given the effort and interest in this aspect of data annotation, the paper would benefit from a more comprehensive description.
3.	The paper could be strengthened by a more thorough analysis of cases where the model's predictions have failed. It would be beneficial to examine instances of high misclassification rates for certain labels and to propose solutions for addressing these challenges.
4.	The requirement for skeleton data as input for the TRANSM3E model may present practical challenges, as skeleton data may not be readily available in all video datasets. The computational overhead associated with extracting this data could impact the model's applicability in real-world scenarios.
5.	The length of Section 4 is disproportionate to the rest of the paper. It is recommended that the experimental section could be a separate section, titled EVALUATION (Section 5).

**Suitability:**

2

---

### Meta-Review · Area_Chair_dwqW · 2024-06-30

**Recommendation:** Accept (Poster)
**Confidence:** 5

**Metareview:**

The paper tackles the novel problem of activated muscle group estimation from in-the-wild exercise videos of High Intensity Interval Training. The primary contributions from the work are a new multi-modal dataset and a new approach for the challenging task.

The paper received reviews which were positive for the most part - 4 weak accept and 3 borderline accept. The rebuttal from the authors convinced two of the reviewers to revise their scores upwards. There were some concerns about the writing quality and to some extent, the methodology and evaluation.

I agree with the assessment of the reviewers. For a first of its kind task, the paper covers reasonable ground and sets the stage for further work on an important problem related to human activity understanding.

**I recommend acceptance of the paper**. The authors should address the concerns raised by the reviewers and improve the quality of final draft. The authors should also share necessary links for downloading the dataset and the codebase of the paper.